# Assessing the Impact of Road Traffic Externalities on Residential Price Values: A Case Study in Madrid, Spain

**DOI:** 10.3390/ijerph16245149

**Published:** 2019-12-17

**Authors:** Francisco Guijarro

**Affiliations:** Research Institute for Pure and Applied Mathematics, Universitat Politècnica de València, 46022 Valencia, Spain; fraguima@upvnet.upv.es; Tel.: +34-96-387-7000

**Keywords:** road traffic, residential price, hedonic model, traffic externalities

## Abstract

This paper describes a study of the relationship between undesired road traffic externalities and residential price values in the Spanish city of Madrid. A large database was gathered, including the price and characteristics of 21,634 flats and road traffic intensity at 3904 different points across the city. The results obtained by a hedonic model suggest that both distance from the traffic measurement point and average daily traffic are significantly related to the price of residential properties, even after controlling for structural and neighbourhood variables. Distance to traffic areas has a positive impact on dwelling prices, whilst these are negatively related to traffic intensity.

## 1. Introduction

According to the European Environment Agency, transport is one of the most significant sources of air pollution, especially that of particulate matter (PM) and nitrogen dioxide (NO${}_{2}$). In fact, greenhouse gas emissions from transport have increased in recent years and average CO${}_{2}$ emissions also rose for the first time in 2017 because of a higher number of cars. Road traffic is reported as the most widespread source of environmental noise in European countries, with around 100 million people exposed to average sound levels of 55 dB or higher during the day, evening and night from road traffic noise only.

Negative externalities from transport have been highlighted in the literature, ranging from sleep disturbance [1,2], increasing risk of stroke [3,4], hyper and hypotension [5,6], depression [7], and others [8,9,10]. These negative externalities can influence the price of residential properties near roads with heavy traffic. The literature has also highlighted the positive externalities associated with proximity to road traffic. Properties close to arterial streets have better access to public transport, urban amenities, etc. The impact of road traffic and its externalities on residential property prices depends upon “the relative influence of traffic’s positive and negative externalities” [11]. As [12] claims, road traffic externalities should be included in regional planning policies. This useful tool would help to determine the improvements that public interventions could generate to benefit property prices and house owners. On the other hand, assessing the decline in values reveals the damage caused by environmental deterioration.

The influence of road traffic on property prices has been extensively analyzed in the literature. The findings are mostly consistent with the perception that the effect of negative externalities outweighs positive externalities on property prices. Under the assumption that negative externalities are capitalized into house values, a hedonic price method was applied by [13] to analyze the impact of traffic noise on the value of 292 housing transactions in Stockholm during the period 1986–1995. The author concluded that properties located near a road were sold at an average discount of 30%. The impact of traffic noise on housing values was also gauged by [14], who gathered information on property rents and characteristics in the city of Geneva (13,034 rented dwellings). The hedonic approach measured the impact of all sources of noise on rental prices, and concluded that a 1% noise increase lowered rents by 0.7%. The price reduction was even greater when only aircraft noise was considered: 1%. The hedonic method also reported a decline of 1.3% in land prices for an increase of traffic noise of 1% in Seoul [15], considering distance to different types of road along with traffic noise intensity.

However, the hedonic method reported different results for three UK cities; road traffic noise had a negative impact on flats in London, no significant impact in Birmingham and a positive one in Sutton Coldfield [16]. The latter case can be partially explained by the limited sample analyzed: 86 flats in Sutton Coldfield, compared with 226 in Birmingham and 407 in London. The relationship between traffic noise and the price of undeveloped properties for house building was analyzed by applying the hedonic method in [17]. The results showed that plots in zones with excess noise were about 57% cheaper than those in quieter areas. A weakness of this analysis was the limited sample used in the research, only 56 properties, a low number considering that other binary variables were included in addition to price (green areas, forest proximity, factory area, electricity, etc.). Another example is the study by [18], who measured the percentage change in property prices per noise level in dB in the Polish city of Olsztyn. The distribution of the prices of apartments was mapped by ordinary kriging. The results show that price differences were explained by sociodemographic characteristics, different locations and noise exposure. Housing price trends in the municipalities around two newly developed highways in the Netherlands were analyzed by [19], who reported that “changes in accessibility result in a significant positive effect on the price of housing in nearby municipalities, but that increased noise pollution and traffic intensity levels result in lower price levels”.

The impact of freeway truck traffic on residential property values was studied by [20], who collected data on 4715 single-family houses in Los Angeles, California. The conclusion was that a 1% increase in truck traffic reduced prices by $2000 to $2750 for a $420,000 house near the freeway. The hedonic model proposed by [21] categorizes independent variables into different groups. The authors included the characteristics of the properties (size, age, rooms, etc.), neighbourhood characteristics (elderly people, income, foreign people, etc.), and accessibility as the control variables. They were thus able to isolate the effect of noise exposure on residential property prices and concluded that price discounts to the amount of 0.23% followed a 1 dB increase in traffic noise in Hamburg. Table 1 summarizes those papers which analyze the impact of traffic externalities on housing prices, and highlights the methodology used by authors, the variables to measure the traffic impact, sample size, cities, and key findings. A thorough review of papers on the impact of noise pollution on real estate values can be found in [12].

From the above references we can conclude that negative externalities have a greater impact on property values than positive externalities. However, the net impact on prices largely depends on the country and city, the control variables used, and type of property studied. Some of the results were obtained from small samples, which prevented the hedonic model from reaching significant conclusions. Fortunately, researchers are focusing on developing mass appraisal methodologies [30,31] when large amounts of information are available, and this should improve their significance.

The aim of this paper was thus to empirically examine the impact of the negative externalities associated with road traffic from a large database for the city of Madrid, Spain. Our assumption was that the higher the daily traffic, the higher the exposure of inhabitants to noise and air pollution (the main negative externalities of road traffic), and this should negatively impact property prices. Our second assumption was that the further the distance from the dwelling to the nearest traffic measurement point, the lower the negative impact on the property price. To the best of the author’s knowledge, no other research has been carried out with such a large number of dwellings and points for traffic measurement. Madrid is the focal point of the most important road communications in the country, as it is also the main commercial centre because of its strategic location in the centre of the country. Most motorways connecting the main cities start from Madrid, its airport is the main connection for both international and domestic flights, and it is also the hub of the high-speed train lines. However, the impact of road traffic on residential price values cannot be analyzed solely by considering prices and road traffic metrics. The key drivers of house prices have been extensively reported in the literature, including the effect of road traffic externalities.

In this study we used several variables related to the property characteristics and the neighbourhood as control variables. The results suggest that distance from the dwelling to its nearest traffic measurement point is positively related to price, whilst the average daily traffic on the road has a negative effect, even after considering structural and neighbourhood characteristics. A significant issue of our research is the large dataset used by combining different sources of information.

The rest of the paper is structured as follows. Section 2 describes the database used plus other sources of information, along with the hedonic used to gauge the impact of traffic. Section 3 discusses the results obtained when applying the methodology to the data from Section 2, and our main conclusions are given in Section 4.

## 2. Materials and Methods

The financial crisis had a negative impact on the number of vehicles in the region of Madrid. Figure 1 shows a drop in the vehicle fleet during that period, which was immediately recovered in the following years. At the end of 2017, when the crisis was hypothetically over, the reported number of vehicles in the whole region showed a large increase. This had an impact on the noise and pollution in Madrid, having regard to the ageing of the vehicle fleet. Both during and after the financial crisis the percentage of old vehicles (10 years and older) has been steadily increasing. In 2017, more than 60% of the vehicles in Madrid were 10 years or older, compared to 36% in 2008. This means that the end of the financial crisis did not give rise to the renewal of the vehicle fleet. Furthermore, most of the old vehicles are not environmentally friendly.

The regional government periodically publishes information regarding the traffic flow and intensity on the main roads in the city (www.madrid.es/portales/munimadrid/es/Inicio/El-Ayuntamiento/Estadistica). This information includes the points at which traffic intensity is measured and the vehicle flow (number of vehicles per hour). It also includes the average speed registered in each TMP and average density of roads in percentage terms. However, these two variables have many missing values which impede its statistical assessment. We registered the physical location of 3904 traffic measurement points (TMP) in Madrid, along with the average daily traffic (ADT) per TMP for the month of July 2019. Average daily traffic was measured as the number of cars passing that point daily. TMPs are mainly located in those roads with heavy traffic, while quiet areas have a lower number of TMPs. Hence, our assumption is that inhabitants will assess a higher value to those properties located in quiet areas, far away from noisy and polluted areas surrounded by heavy traffic roads, where buyers are not willing to pay higher prices for similar properties in quiet areas.

The situation of the TMPs is given in Figure 2. The picture highlights the main roads across Madrid, and how most TMPs are in certain main roads to better control the traffic flow. However, other TMPs are positioned on minor roads, so that the picture gives a panoramic view of the traffic density in Madrid. The red lines depict the boundary of Madrid city.

To examine the abovementioned assumptions, we collected information from 21,634 dwellings until the end of July 2019 from a popular real estate website (Idealista.com) which offers more than 1.7 million properties all over Spain, including 30,000 in Madrid. We recorded the following variables per dwelling: price, area, number of bedrooms, number of bathrooms, address, and whether the building had a lift. The locations can be seen in Figure 3.

The website also includes information on the dwelling given by the seller. However, this was in the form of an unstructured text, which made it difficult to analyze and extract relevant information regarding the market value of the property. A text mining algorithm was thus encoded in R software [32] to extract useful variables related to prices (R version 3.6.1). The algorithm computed the following binary variables: garden, renovated, views, luxury, open kitchen and wooden floor. For example, the following is an extract of the description of a flat in the centre of the city: “*Luminoso y con amplias estancias. Cuarta planta exterior reformado para entrar a vivir*” (Light and spacious. Located in the fourth floor, exterior renovated flat ready to start living in). The word “*reformado*” translates into renovated, which scores a value of 1 in the corresponding binary variable. Another example: “*Amplia vivienda situada en una décima planta con mucha luz y unas vistas impresionantes del Skyline de Madrid*” (Spacious property located on the 10th floor, very luminous and stunning views of Madrid Skyline), in which “*vistas impresionantes*” translates into stunning views.

The above-mentioned variables were the group of structural variables and were used to design the hedonic model that explained property prices. The hedonic price theory assumes that each house has its own unique attributes. These characteristics are not traded explicitly, and hence their prices cannot be observed on the market. The hedonic pricing model is applied to estimate the marginal contribution of each property characteristic to the property price. The assumption of any hedonic price model is that there is a function, hedonic, that determines the price of all goods in the market, where $price_{i}=hedonic\left(attribute_{i}\right)$, $price_{i}$ is the price of the property *i*, and $attribute_{i}$ is the vector with the characteristics of property *i*. Although location was included, we decided to consider an additional factor related to the neighbourhood characteristics. Madrid is divided into census sections, which is the minimal administrative cluster to which all properties are assigned. This means properties in the same census section share some common characteristics and their prices follow a similar pattern, different to those in other census sections. Location has traditionally been considered a potential driver of sales prices in the real estate market, and this has led researchers to propose different approaches to consider the spatial effect on housing prices [33,34,35,36].

Each dwelling in the dataset was assigned to its census section according to the coordinates downloaded from the Spanish Statistical Office’ website (www.ine.es). The latest information on the average family income per census section, which differed widely between different sections, was also included in the dataset.

Figure 4 shows the average price per square metre in the different census sections. These were computed by considering the dwellings in each section. The figure shows higher prices concentrated in the city centre, to the north of the popular El Retiro park, where prices rise to €9000 per square metre. The grey areas have no dwellings (mainly green and industrial areas).

The last group of variables considered in the study was the distance from each dwelling to the nearest TMP (km), plus its average daily traffic count (number of vehicles). This traffic intensity was also added to the dwelling. A summary of the main statistics of the involved variables is given in Table 2. Although more dwellings were on offer on the real estate website, we excluded all those whose address was not given.

We followed the standard hedonic model to analyze the impact of road traffic on house prices. The sales price of properties was modelled from their structural, neighbourhood, and road traffic variables. This showed the effect of road traffic on the property prices after controlling the variables commonly related to price. The hedonic model is presented in Equation (Equation 1). Log(Price) is the natural logarithm of the sales price, α is the intercept term of the regression, $\beta_{i}$ are the estimated coefficients for the independent variables and ϵ is the error term.
(1)$$\begin{array}{c} log\left(Price\right)=\alpha +\beta_{1}Area+\beta_{2}Bedrooms+\beta_{3}Bathrooms+\beta_{4}Lift+\beta_{5}Avg\_family\_income+\\ \beta_{6}Renovated+\beta_{7}Views+\beta_{8}Luxury+\beta_{9}Open\_kitchen+\\ \beta_{10}Wooden\_floor+\beta_{11}Distance+\beta_{12}ADT+\epsilon \end{array}$$

We considered the log of price because this variable is usually skewed to the right, which translates into greater importance for the more expensive dwellings in the regression model. In order to avoid this situation, most researchers when dealing with prices use the logarithm transformation [11,17,20,24,27,28,37,38]. This transformation also gives a different interpretation of the estimated coefficients, so that the relative impact on prices for a unit change in the independent variables can be computed.

## 3. Empirical Results

The model results are presented in Table 3. We can see that the model explains 94.35% of the variation in dwelling prices, according to the adjusted $R^{2}$, which is consistent with similar results in large cities. The F-statistic is highly significant, with 12 and 21,621 degrees of freedom, because of the large size of the sample.

The estimated coefficients have different statistical significance. The variables with significant coefficients at the 99.9% confidence level are area: number of bathrooms, lift, average family income, renovated, and views. All the coefficients except open kitchen are positive, as expected. Having an open kitchen reduces the estimated price of the dwelling, but this is explained by the positive significant correlation between the surface area of the dwelling and the presence/absence of an open kitchen (+0.237). Previous studies have reported similar cases, with the sign of some coefficients being opposite to the sign of the correlation between variables [39]. The non-significance of the coefficient associated with the number of bedrooms was for a similar reason. This variable is also closely correlated with the floor area (+0.702), so that a high p-value is obtained because the effect of extra bedrooms is also included in the floor area. In other words, once the model has considered the surface area, the number of bedrooms has no effect on price variations. However, the number of bathrooms is significant. In our view, this is because this variable acts as a proxy variable for the age of the building, although we were not able to include it in the dataset because it was omitted from the real estate website. In Spain, most old dwellings only have one bathroom, while those 20 years old or less have at least two bathrooms/toilets, regardless of their size. We assume that the number of bathrooms is included in the model, not because it explains the size (correlation with area), but because of the relationship between age and number of bathrooms.

A wooden floor and distance from the dwelling to the nearest TMP are significant at the 99% confidence level. The average daily traffic (ADT) gets a significant coefficient at the 95% confidence level. We can thus conclude that property prices are positively related to distance from the nearest TMP, even after accounting for other relevant variables of price variations. A similar conclusion can be reached regarding the level of road traffic. The ADT variable is also significant in the regression model, even though its associated confidence level is less than that reported for distance to the TMP.

Luxury dwellings are of course expected to be more expensive than non-luxury properties. However, the luxury coefficient was curiously non-significant in the model. Our hypothesis is that sellers use this adjective as a publicity stunt, but the quality of the dwelling is not necessarily connected to the use of this adjective. Nevertheless, the coefficients associated with binary variables should be carefully analyzed. Their non-significance can be partially explained by the unbalanced data. For example, only 4% of the dwellings are declared as luxury, and only 7% have an open kitchen. The lower the percentage of any class, the higher the influence of those dwellings with a value of 1 in the corresponding variable. The significance of the variable could thus ultimately be determined by a small bundle of dwellings, and this can translate into a biased analysis of the variable importance.

From Table 3 we can conclude that our findings are aligned with those of other studies reported in the literature. Negative externalities have a statistically significant impact on house prices, but it is almost negligible when compared with the influence of structural and neighbourhood variables. The impact of distance from the dwelling to the nearest TMP has a similar significance to the average daily traffic registered at that point, according to the estimates in Table 3.

The non-significance of the bedrooms’ coefficient could be explained because the relation between the area and the number of bedrooms. We have performed a multicollinearity diagnostics to confirm whether the regression model was affected by the high correlation between some variables. The variance inflation factor (VIF) was applied on results from Table 3. The highest VIF was reported for the variable Area, with an estimate of 4.6—followed by Bathrooms with a value of 2.8.—According to [40], multicollinearity is high when VIF is greater than 10. This way, we can conclude that the regression results are not affected by multicollinearity.

A limitation of our work is that road traffic was the only source of noise considered in the research. Further analysis should be carried out to include other externalities such as rail or aircraft noise. Another interesting approach could investigate whether diurnal noise levels can impact housing prices in a different way to nocturnal noise levels.

## 4. Conclusions

Although residential properties near busy main roads can benefit from different advantages, they can also be affected by negative externalities. Our research analyzed the impact of two traffic-related variables on residential property prices: distance to the nearest traffic measurement point and the average daily traffic registered at that point. The hedonic model considered different structural and neighbourhood variables related to price. The impact of traffic on price was computed by accounting for these traditionally considered variables in the appraisal context. The empirical analysis was performed on a large dataset composed of properties in Madrid. The characteristics and prices of these properties were gathered from a leading real estate website, and included variables such as area, number of bedrooms, etc., and others extracted from the description of the property (views, renovated, etc.). The dataset was completed with information from 3,904 traffic measurement points.

Our findings suggest that both of the traffic-related variables can influence residential property prices, and that the distance between a dwelling and the nearest traffic measurement point has a positive impact on its price, while this is negatively affected by the average daily traffic. It is interesting to compare these results with those of other studies on the subject. Some papers that relate real estate values to noise pollution (one of the negatives externalities of living near main roads) reported a price reduction of between 0.16% and 1.45%. We also found that the impact of these variables on price is limited when other variables are considered in the explanation of price dispersion. An extra 1 kilometre in the distance to a TMP translates into an average price increase of 0.7%, while every 100,000 vehicles passing a TMP reduce prices by 1%.

## Figures and Tables

**Figure 1 ijerph-16-05149-f001:**
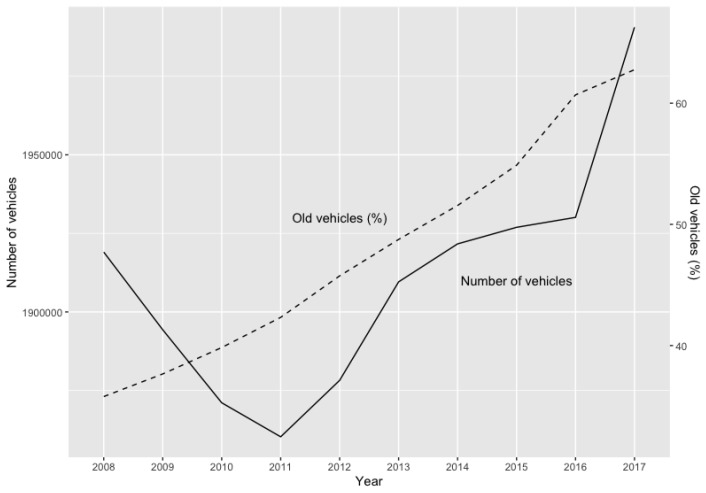
Evolution of vehicles in the region of Madrid. Source: National Department of Traffic, www.dgt.es.

**Figure 2 ijerph-16-05149-f002:**
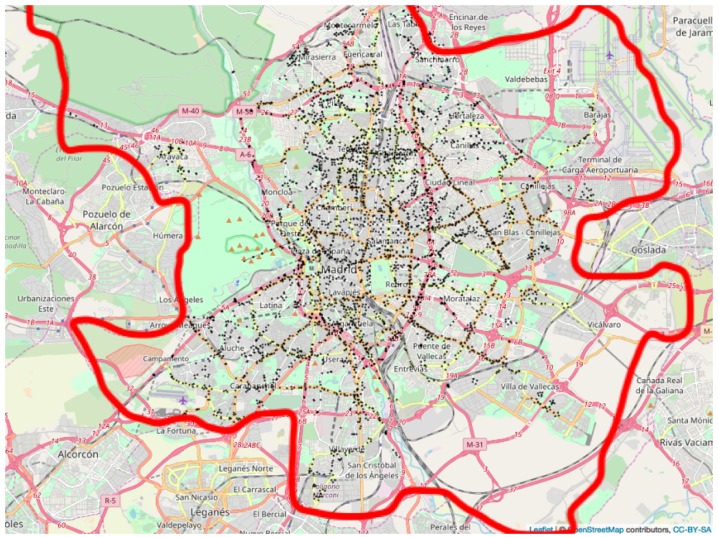
Location of traffic flow measurement points across Madrid.

**Figure 3 ijerph-16-05149-f003:**
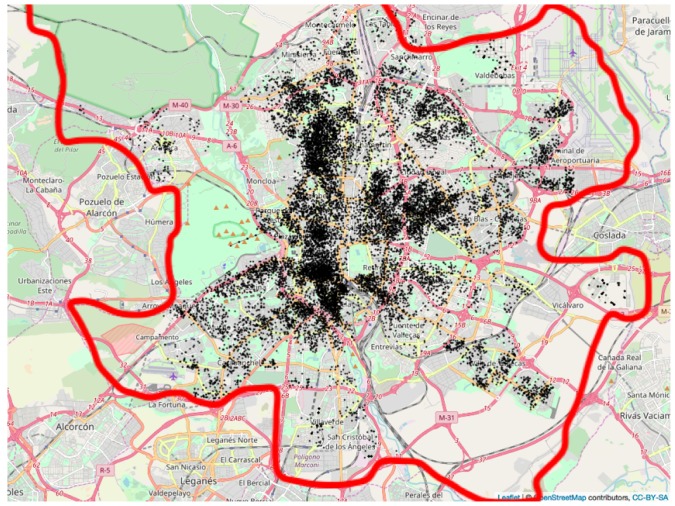
Location of flats in the analysis.

**Figure 4 ijerph-16-05149-f004:**
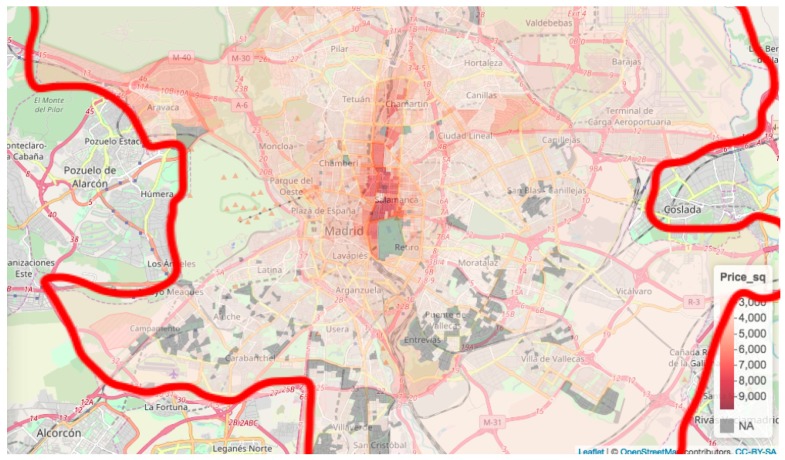
Average price of flats per square metre by census section.

**Table 1 ijerph-16-05149-t001:** Summary of papers that investigate external costs of traffic.

Paper	Quantitative Measurement	Methodology	Database Size	City	Key Results
Wilhelmsson [13]	Noise level, visual exposure to a road	OLS	292 housing transactions	Stockholm (Sweden)	A single-family house located near a road with loud traffic could lose 30% of its value
Kawamura and Mahajan [22]	Daily volume, max hourly volume, and total night volume for trucks and all vehicles	Spatial lag and 2SLS	685 single-family houses	Chicago (USA)	Traffic characteristics have modest but statistically significant impact on property values; truck traffic characteristics are not statistically significant
Day et al. [23]	Road, rail, and aircraft noise	Partially linear model with spatial smoothing	10,848 residential properties	Birmingham (UK)	A 1-dB increase in road traffic noise reduces the selling price of a property by between 0.18% and 0.55%
Kim et al. [15]	Traffic noise level, distance to different types of roads	OLS	328 zones	Seoul (South Korea)	A 1% increase in traffic noise is associated with a 1.3% decline in land price
Andersson et al. [24]	Road and rail noise	OLS	1738 houses	Lerum (Sweden)	Road noise has a larger negative impact on the property prices than railway noise
Blanco and Flindell [16]	Road and rail noise	OLS	Flats in London (407), Birmingham (226) and Sutton Coldfield (86)	London, Birmingham, Sutton Coldfield (UK)	Noise decreases property values in London, raises in Sutton Coldfield, and no significant impacts in Birmingham
Brandt and Maennig [21]	Road, rail, and aircraft noise on a micro-level grid	Spatial lag	4832 condominiums	Hamburg (Germany)	A 1-dB(A) increase in noise reduces the value of a condominium by 0.23%
Larsen [25]	Daily traffic count	OLS	9680 single-family house transactions	Kettering (USA)	Parcels fronting or adjacent to a high-traffic sell at an 8.1% discount
Li and Saphores [20]	Freeway average daily traffic, percentage of trucks in freeway	OLS	4715 single-family houses	Los Angeles (USA)	A 1% increase in total traffic would reduce by only $24 the value of a $420,000 house located within 100 m of a freeway. A 1% increase in truck traffic would decrease the value of a $420,000 house located between 100 and 400 m from the nearest freeway by $2000 to $2750
Del Giudice and De Paola [26]	Noise level	OLS	31 residential properties	Naples (Italy)	A 1-dB increase in noise level reduces the selling price of a property by between 0.30% (diurnal emissions) and 0.33% (nocturnal emissions)
Larsen and Blair [11]	Daily traffic count	OLS	9680 single-family house transactions and 455 multi-unit rental properties	Kettering (USA)	Houses located adjacent to an arterial street sold at a 7.8% discount
owicki and Piotrowska [17]	Noise level, distance to roads	OLS	56 residential properties	Poznan (Poland)	Plots located in the zone with noise exceedance at night were about 57% cheaper than those located outside this zone
Szczepańska et al. [18]	Noise level	Correlation analysis	118 apartments	Olsztyn (Poland)	Correlation between apartment prices and noise level in the range [0.61, 0.51]
Swoboda et al. [27]	Traffic noise	LWR (locally weighted regression)	42,083 single-family properties	St Paul, Minnesota (USA)	Marginal effect of traffic noise varies over space and time
Le Boennec and Salladarré [28]	Air and noise pollution	OLS	2969 houses	Nantes (France)	Air pollution has no significant impact on the price; noise pollution does have an impact
Gallo [29]	Average daily frequency	OLS	60 zones	Naples (Italy)	High-frequency metro lines have appreciable effects on real estate values

**Table 2 ijerph-16-05149-t002:** Summary statistics for the variables included in the dataset.

Variable	Description	Min	Median	Mean	Max	Sd. Deviation
Structural variables						
Price	Bid price (€)	99,500	258,860	377,273	3,600,000	335,569.80
Area	Square metres of the dwelling	30	87	100.30	300	51.65
Bedrooms	Number of bedrooms	1	3	2.62	8	1.15
Bathrooms	Number of bathrooms	1	1	1.62	5	0.08
Lift	Lift facility in building (0/1)	0	1	0.76	1	0.43
Garden	Garden (0/1)	0	0	0.20	1	0.41
Renovated	Is the dwelling renovated? (0/1)	0	0	0.24	1	0.43
Views	Dwelling with views (0/1)	0	0	0.14	1	0.35
Luxury	Luxury dwelling (0/1)	0	0	0.04	1	0.20
Open kitchen	Open kitchen (0/1)	0	0	0.07	1	0.26
Wooden floor	Wooden floor / laminate flooring (0/1)	0	0	0.27	1	0.45
Neighborhood variable						
Avg. family income	Average family income in the section (€)	10,027	34,161	39,626	89,015	5508.44
Road traffic variables						
Distance	Distance from the nearest TMP (km)	0.0003	0.1159	0.1983	4.047	0.36
ADT	Average daily traffic in the nearest TMP	0.5	19,800.9	31,783.6	345,239.4	37,471.69

**Table 3 ijerph-16-05149-t003:** Regression results.

Variable	Estimate	Std. Error	t Value	Pr(>|t|)	VIF
(Intercept)	11.2820040	0.0040825	2763.48	0.0000 ***	
Area	0.0103689	0.0000416	249.14	0.0000 ***	4.6157
Bedrooms	−0.0013296	0.0012373	−1.07	0.2826	2.0577
Bathrooms	0.0239193	0.0021622	11.06	0.0000 ***	2.8575
Lift	0.0429986	0.0025190	17.07	0.0000 ***	1.2008
Avg. family income	0.0000133	0.0000002	57.26	0.0000 ***	1.6408
Renovated	0.0110624	0.0023904	4.63	0.0000 ***	1.0520
Views	0.0121576	0.0029353	4.14	0.0000 ***	1.0474
Luxury	0.0029306	0.0052594	0.56	0.5774	1.0814
Open kitchen	−0.0149898	0.0039775	−3.77	0.0002 ***	1.0987
Wooden floor	0.0066911	0.0022680	2.95	0.0032 **	1.0326
Distance	0.0076939	0.0027497	2.80	0.0051 **	1.0058
ADT	−0.0000001	0.0000000	−2.03	0.0422 *	1.0059

Significant codes: 0.001 ***, 0.01 **, 0.05 *; Residual standard error: 0.1472 on 21621 degrees of freedom; Multiple $R^{2}$: 0.9437; Adjusted $R^{2}$: 0.9437; F-statistic: 3.022 × 10${}^{4}$ on 12 and 21621 DF, *p*-value: < 2.2 × 10${}^{-16}$.

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
