# Peer review of "Assessing the Impact of Road Traffic Externalities on Residential Price Values: A Case Study in Madrid, Spain"

_ijerph, 2019, doi:10.3390/ijerph16245149_

Round 1
Reviewer 1 Report
Assessing the Impact of Road Traffic Externalities on Residential Price Values: a Case Study in Madrid, Spain.
This paper analyzes the relationship between road traffic on housing prices in Madrid (Spain). This paper is an application of ordinary least square (OLS) to estimate a hedonic model. I think that the data set is well, the method used is the classic OLS without consider spatial effects and the idea is good, but I have some comments:
1. I think that the model has multicollinearity problems. The author has not checked the multicollinearity in the model. The authors should show some test about this. Is possible that the sign of "Open kitchen" and that the "Bedrooms" variable is not significative is because there is multicollinearity. In fact, the authors say, in line 170, that the correlation coefficient is 0.702. It is likely that in this model exist a severe multicollinearity problem.
2. The Distance variable is "the distance from each dwelling to the nearest TMP". Having a TMP near to the house has a positive or negative impact on houses? Why?
Why do you think that the ADT variable has a negative effect on housing prices?. I think that the author should clear this issues.
3. If the sign of coefficient of Distance is positive, why the author say that a extra km causes a price reduction (line 218)?
4. lines 124-127. The author should translate into English language the Spanish phrases
5. Although the author doesn't consider the spatial effect on housing prices, he could include some references about that (Dubin, 1992; Chica-Olmo, 1995; Bourassa, Cantoni, & Hoesli, 2010; Ibeas, Cordera, et al, 2012)
Author Response
Please find the revised version of the manuscript and my point-by-point answers to the reviewer's suggestions.

Reviewer 2 Report
The aim of the paper is to study the relationship between undesired road traffic externalities and residential market values in the Spanish city of Madrid. Although the issue is of considerable interest and relevance, there are some shortcomings in the paper, so major revisions are needed. In particular, it’s suggested to:
improve the content of paragraph 1"Introduction" regarding studies in the literature that addressed the relationship between undesired road traffic externalities and residential market values, possibly by entering a summary table, depending on: quantitative measurement (e.g. number of cars in an hour) or qualitative (e.g. score assigned by panel of experts; questionnaires administered to inhabitants etc.) of road traffic externalities; methodological approaches and evaluation techniques used; number of apartments in the database considered by the analysis that used the hedonic price method for the same purpose as this work; size of the cities analyzed and their population density affecting the level of traffic; key findings of previous studies on the city of Madrid (rows 76-80).The paragraph of the introduction should allow to have a general view of the state of the art of the road traffic externalities and residential market values, emphasizing the controversial and diverging hypotheses or gaps in literature;
indicate the actual number of the study sample by specifying whether it consists of 21,534 (row 3 in the Abstract section) or 21,634 flats (row 114 in the Material and method section); enter in paragraph 2 "Materials and methods" the source of data on the number of vehicles of the entire region of Madrid analyzed and a brief nod to the trend of the number of vehicles for the city of Madrid alone, from 2008 to 2017, in order to allow immediate comparison with subsequent analyses carried out; insert in paragraph 2 “Materials and methods” a briefly description about the theory assumption of the hedonic price method adopted, and the names and versions of any software used for the application (e.g. R software, row 121); briefly specify whether the location of traffic measurement points is located at specific points in the city congested from traffic in relation to the second assumption that the further the distance from the nearest dwelling to the nearest traffic measurement points (TMP), the lower the negative impact on the property price (rows 107-110); move the part relating to the two assumptions that this study aims to verify (rows 106-109), in the final part of paragraph 1 linking them to the objective of the work and the expected results in order to return a clear treatment of the hypotheses that the aim of the work would like to verify; mark in Figure 2, 3 and 4 the delimitation of the area where the database properties are located in order to highlight the territorial boundaries of the study area; return in graphic form, possibly the same as adopted for Figures 2, 3 and 4, the ranges of the traffic level detected by the TMPs for the different areas of the city in order to be able to verify the consistency of the results obtained in the different areas affected by traffic; move Table 2 relating to the regression results so that it is closer to the comments in paragraph "3. Empirical results" in order to allow a direct comparison of the results obtained; highlight the limitations of the work and the future developments of this research at the end of paragraph 3 to allow others to build on published results; specify the measurement approach used to quantify the distance variable from the nearest TMP (e.g. km on foot, by cars etc.) in order to return a clearer understanding of the collected data; verify that the results are in line with the expected empirical evidence for road traffic externalities such as "Distance from the nearest TMP point" and "Average daily traffic". Specifically, the estimated coefficient value returned by the model (0.0051 for the Distance variable and 0.0422 for the ADT variable) and especially the coefficient associated with the ADT variable (equal to -0.0000001) returns a marginal contribution almost null on the dependent variable of the price. In addition, the positive functional link of the variable "Distance from the nearest TMP point" with the price shows that as the distance grows from the nearest TMP point, this will determine an increase for the price by 0.7%, contrary to what is exposed in the lines 217-219. It is also suggested that both Fisher's F values and VIF values for independent regression model variables should be reported in Table 2.Author Response
Please find the revised version of the manuscript and my point-by-point answers to the reviewer's suggestions.

Round 2
Reviewer 1 Report
I reckon that the paper has been improved and it can be published.
Reviewer 2 Report
The main indications have been taken into account. The paper could be published.